# Finite Symmetries in Agent-Based Epidemic Models

**Gilberto M. Nakamura** [1,2] , **Ana Carolina P. Monteiro** [1] , **George C. Cardoso** [1]
**and Alexandre S. Martinez** [1,2,*]

[1]  Faculdade de Filosofia, Ciências e Letras de Ribeirão Preto (FFCLRP), Universidade de São Paulo (USP),
    Avenida Bandeirantes 3900, 14040-901 Ribeirão Preto, São Paulo, Brazil; gmnakamura@usp.br (G.M.N.);
    ana.carolina.monteiro@usp.br (A.C.P.M.); gcc@usp.br (G.C.C.)

[2]  Instituto Nacional de Ciência e Tecnologia de Sistemas Complexos (INCT-SC), Rua Dr. Xavier Sigaud 150,
    Urca, 22290-180 Rio de Janeiro, Brazil

*  Correspondence: asmartinez@usp.br; Tel.: +55-16-3315-3720

**Abstract:** Predictive analysis of epidemics often depends on the initial conditions of the outbreak, the structure of the afflicted population, and population size. However, disease outbreaks are subjected to fluctuations that may shape the spreading process. Agent-based epidemic models mitigate the issue by using a transition matrix which replicates stochastic effects observed in real epidemics. They have met considerable numerical success to simulate small scale epidemics. The problem grows exponentially with population size, reducing the usability of agent-based models for large scale epidemics. Here, we present an algorithm that explores permutation symmetries to enhance the computational performance of agent-based epidemic models. Our findings bound the stochastic process to a single eigenvalue sector, scaling down the dimension of the transition matrix to $o(N^2)$.

**Keywords:** Markov processes; computational methods; epidemic models; complex systems; nonlinear dynamics

## 1. Introduction

In recent years, the emergence of Zika and Ebola viruses have attracted much attention from the scientific community after reports of their aggressive effects, respectively, microcephaly in newborns [1] and high mortality rate [2–4]. Despite their intrinsic differences concerning transmission mechanisms and pathogen-host interaction, both viruses spread in a population starting from a few infected individuals, based on their geographic location and network of contacts. Contact tracing and proper clinical care planning are critical parts of the WHO strategic plan [5] to mitigate ongoing transmissions and incidence cases, requiring the correct spatiotemporal dissemination of the disease. This assertion has renewed the interest in agent-based epidemic models (ABEM).

ABEM are mathematical models that describe the evolution of infectious diseases among a finite number $N$ of agents over time (see Reference [6] for an extensive review). For that purpose, agents are labeled using integer numbers $k = 0, 1, \ldots, N-1$, whereas contacts between agents are mapped via an adjacency matrix $A$. The matrix elements are $A_{ij} = 1$ if the $j$-th agent connects to the $i$-th agent and otherwise vanishes. Accordingly, the set formed by agents and their interconnection is expressed as a graph, as depicted in Figure 1. In this way, heterogeneity arises naturally since the individuality of agents is taken into account, distinguishing ABEM from compartmental epidemic models [7–9].

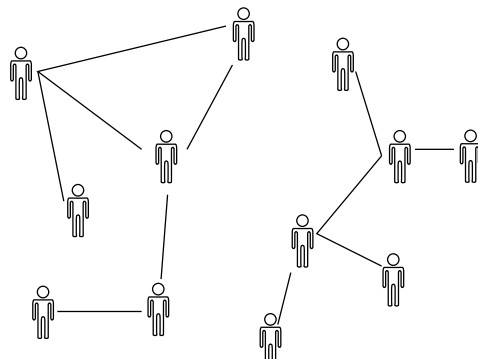

**Figure 1.** Agent network. Agents (vertices) and their interconnections (edges) are expressed as a graph. The graph representation introduces heterogeneity among the agents, which must be accounted for during disease spreading.

The susceptible-infected-susceptible model (SIS) is the simplest ABEM. It considers only two health states for agents, infected $|1\rangle$ or susceptible $|0\rangle$, and the occurrence of the following events during a time interval $\delta t$ [10,11]. An infected agent may undergo a recovery event and return to susceptible state with probability $\gamma$; an infected agent may infect a susceptible agent with transmission probability $\beta$ if and only if both agents are connected; or remains unchanged, as Figure 2 illustrates. Therefore, the SIS ABEM is inherently a Markov process in discrete time. The time interval $\delta t$ is often chosen so that sequential recovery-recovery or transmission-recovery events are unlikely within the available time window. This is the so-called Poissonian hypothesis [12–15].

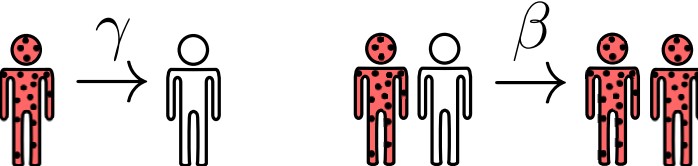

**Figure 2.** Susceptible-infected-susceptible model (SIS) transition events. Infected agents (red dotted) undergo recovery events with probability $\gamma$ and change to susceptible (empty) health status. Infected agents may also infect additional susceptible agents with probability $\beta$, as long they are connected.

Following Reference [16], any configuration of $N$ agents is obtained by direct composition of individual agent states. Let $\mu$ be an integer that labels the $\mu$-th configuration so that:

$$|\mu\rangle \equiv |n_{N-1}\cdots n_1\, n_0\rangle ,\qquad(1)$$

with $n_k = 0, 1$ and $\mu = n_{N-1}2^{N-1} + \cdots + n_0 2^0$. A simple example for $N = 4$ is $|8\rangle \equiv |1000\rangle$, which represents the configuration where only the agent $k = 3$ is infected. From this scheme, it is already clear that there exist $2^N$ configurations in total since there are two available states for each agent. In what follows, we employ the notation: Latin indices enumerate agents $0, 1, \ldots, N-1$, while Greek indices enumerate configuration states $0, 1, \ldots, 2^N - 1$.

Let $|\pi(t)\rangle$ be the probability vector and $\pi_\mu(t) = \langle\mu|\pi(t)\rangle$ the probability of observing the configuration $|\mu\rangle$ at time $t$ [17,18]. The master equation for the general Markov process reads:

$$\frac{d}{dt}\pi_\mu(t) = -\sum_\nu H_{\mu\nu}\pi_\nu(t),\qquad(2)$$

$\hat{H} = (\mathbb{1} - \hat{T})/\delta t$ is the step operator, whereas $\hat{T}$ stands for the transition matrix [19]. The transition matrix $\hat{T}$ encodes all transitions available between any two configuration vectors. Its matrix elements $T_{\mu\nu}$ are constructed from much simpler rules. These rules are model dependent and fully characterize the stochastic model, as we shall see in details later. If a representation of $\hat{T}$ is known, the solution

of the master equation provides the instantaneous values of the probabilities $\pi_\mu(t)$, i.e., the entire probability distribution function. Therefore, it becomes possible to calculate any relevant statistics of the problem at any instant of time, including those that may not be easily accessible or accurate by other numerical methods, such as the instantaneous Shannon entropy or the characteristic function.

Luckily, for time independent $\hat{T}$, the solution is well known:

$$|\pi(t)\rangle = \mathrm{e}^{-\hat{H}t}|\pi(0)\rangle \ . \tag{3}$$

Despite the existence of this exact solution, the applicability of Equation (3) at this stage is limited to small population sizes $N \sim O(20)$. The reason is the exponential growth of the underlying vector space with $N$. Here, we present algorithms to generate the operators $\hat{T}$ and $\hat{H}$ using finite symmetries or, equivalently, permutation symmetries via Cayley's theorem [20]. These algorithms are usually applied to condensate matter physics [21,22], but they may also be employed in epidemiology studies, due to recent developments in the disease spreading dynamics [16]. For pedagogical reasons, we first show how to build the complete $2^N$ vector space and the corresponding transition matrix. Next, we explore cyclic permutations to construct the cyclic vector space, in which $\hat{T}$ is broken down into $N$ smaller blocks. Lastly, we consider the most symmetric cases, which reduce the problem to O($N$). These instances correspond to the mean field or averaged networks. The iteration of sparse $\hat{T}$ over $|\pi(t)\rangle$ produces the desired disease evolution among agents. Relevant steps are shown in Algorithm A1..

## 2. Transition Matrix

The transition matrix $\hat{T}$ for an SIS model considering $N$ two-state agents is [16]:

$$\hat{T} = \mathbb{1} - \beta \sum_{kj} \left[ A_{jk}(1 - \hat{n}_j - \hat{\sigma}_j^+) + \Gamma \delta_{kj}(1 - \hat{\sigma}_j^-) \right] \hat{n}_k \ , \tag{4}$$

where $\Gamma = \gamma/\beta$, $\delta_{kl}$ is the Kronecker delta, $\hat{n}_k|n_k\rangle = n_k|n_k\rangle$ , is the local number operator ($n_k = 0, 1$), and $\hat{\sigma}_k^+|n_k\rangle = \delta_{n_k,0}|1_k\rangle$ , $\hat{\sigma}_k^-|n_k\rangle = \delta_{n_k,1}|0_k\rangle$ are the Pauli raising and lowering local operators, respectively. Local algebraic relationships are $[\hat{n}_k, \hat{\sigma}_{kl}^\pm] = \pm\delta_{klk}$ and $[\hat{\sigma}_k^+, \hat{\sigma}_{kl}^-] = \delta_{klk}(2\hat{n}_k - \mathbb{1})$. Inspection of Equation (4) readily shows $\hat{T}$ is not Hermitian. This means left- and right-eigenvectors are not related by Hermitian conjugation. In this scenario, the correct time evolution of $\pi_\mu(t)$ using Equation (3) requires the complete eigendecomposition, i.e., $2^N$ eigenvalues accompanied by $2^N$ right-eigenvectors and $2^N$ left-eigenvectors. This is often the main criticism against ABEM [12].

However, the scenario described above is not entirely correct. The rationale behind it assumes all eigenstates are equally relevant, which is incorrect whenever $A$ exhibits invariance upon the action of a particular group (sets of transformations). Symmetries allow the matrix representation of $\hat{T}$ to be in block diagonal form, as depicted in Figure 3. Eigenvectors related to each block share the same eigenvalue (degeneracy), as usual in quantum mechanics [23]. Therefore, the trick to simplify problems involving the transition matrix lies in the selection of the appropriate basis in respect to a given symmetry, creating matrix representations with smaller blocks. The computational performance using this methodology surpasses that of working with the full matrix because each block can be treated separately, reducing memory storage and access. In particular, if only a few blocks are relevant to the analysis, the remaining blocks can be neglected. This property often produces massive reductions in the typical dimensions of the problem, enhancing computation times.

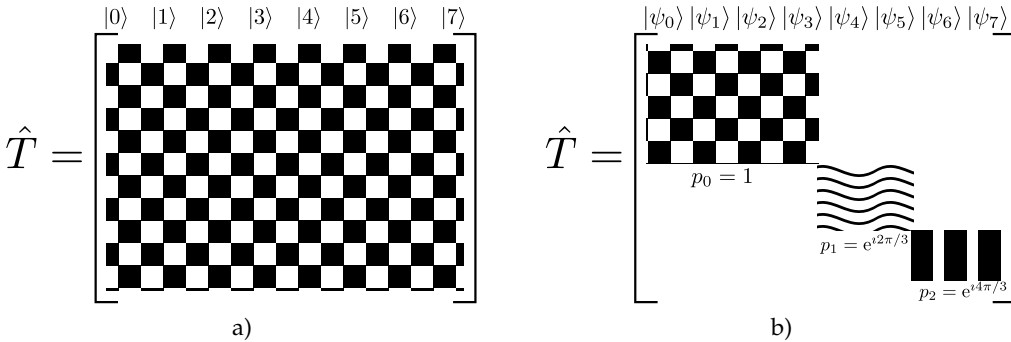

**Figure 3.** Reduction of the transition matrix to block diagonal form. (**a**) In the configurational vector space, $\{|\mu\rangle\}$, the matrix representation of $\hat{T}$ lacks an explicit mathematical pattern. (**b**) The emergence of organizational patterns are observed whenever symmetries of $\hat{T}$ are correctly addressed by employing the eigenvectors $\{\psi\}$ and eigenvalues $\{\lambda\}$ corresponding to the symmetry group considered. Under the invariant basis $\{\psi\}$, the matrix representation of $\hat{T}$ is brought to a block diagonal form, with blocks labeled by eigenvalues $\{\lambda\}$.

In the SIS model, recovery events result from actions of one-body operators, $\hat{\sigma}_k^- \hat{n}_k \equiv \hat{\sigma}_k^-$, on configuration vectors. Infection events are two-body operators: one infected agent may transmit the communicable disease to a susceptible agent after interaction between them, in the time interval $\delta t$. Interestingly, the resulting interaction also depends on symmetries available to the adjacency matrix $A$. The symmetries available to $A$ may be further explored to assemble the initial vector space, reducing $\hat{T}$ to its block diagonal form.

Group operations over $A$ are always finite transformations. One may explore the isomorphism between finite groups and the permutation group via Cayley theorem [20] to build permutation invariant subspaces. To that end, one must select the finite group and the corresponding symmetry. For graphs, the circular representation provides a convenient context to explore the existing symmetries, as Figure 4a depicts. From Figure 4b, connections among agents remain unchanged after cyclic permutation of agents, hence, $A$ exhibits invariance under cyclic permutations. Cyclic permutations form a subset of permutation group and often represent geometric transformations, such as rotations and translations.

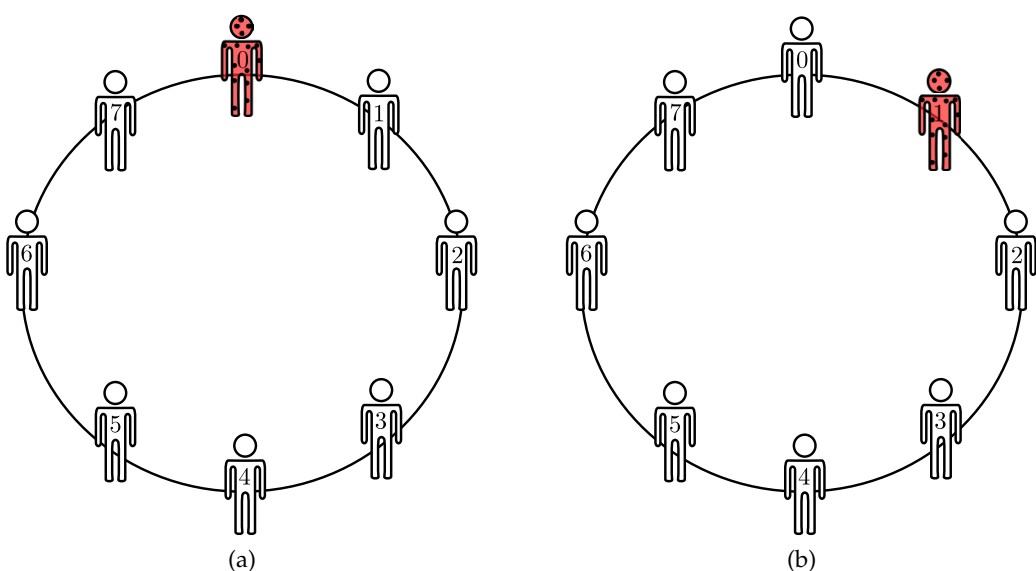

**Figure 4.** Regular graph in circular representation for $N = 8$ and single infected agent (red dotted). (**a**) The infected agent lies at node $k = 0$. (**b**) Graph obtained from cyclic permutation of nodes $k \to k+1$ and $N - 1 \to 0$. Connections remain unchanged.

Vectors with $N$ agents and invariant by cyclic permutations are built as follows. Consider the *representative* vector:

$$|\mu_p\rangle \equiv \frac{1}{\mathcal{N}_\mu} \sum_{k=0}^{N-1} \left( e^{2i\pi p/N} \hat{P} \right)^k |\mu\rangle, \tag{5}$$

where $\mathcal{N}_\mu$ is the normalization and $\hat{P}$ is the single step cyclic permutation with $p = 0, 1, \ldots, N-1$. The eigenvalues $e^{-2i\pi p/N}$ are derived from $\hat{P}^N = \mathbb{1}$. Eigenvalues can be associated with invariant subspaces, or sectors, spanned by their corresponding eigenvectors. For the sake of convenience, the integer $p$ labels the eigenvalue sector. The representative vector $|\mu_p\rangle$ describes the linear combination of $N$-agent configurations related to $|\mu\rangle$ by cyclic permutations. For instance, $|3_0\rangle = (|011\rangle + |110\rangle + |101\rangle) / \sqrt{3}$ corresponds to the representative vector for $\mu = 3$, with $N = 3$ in the $p = 0$ sector. By construction, the vectors $|\mu_p\rangle$ satisfy the eigenvalue equation $\hat{P}|\mu_p\rangle = e^{-2i\pi p/N}|\mu_p\rangle$. They are also useful to identify symmetries, as they never change link distributions, only node labels. If $\hat{T}$ is symmetric under cyclic permutations, $\hat{T}$ and $\hat{P}$ commute with each other $[\hat{T}, \hat{P}] = 0$, meaning they share a common set of eigenvectors. Thus, $\hat{T}$ can be written using $|\mu_p\rangle$ and, more importantly, transitions between eigenvectors with distinct eigenvalues are prohibited. This feature leads to a block diagonal form to the matrix representation of $\hat{T}$.

## 3. Cyclic Vector Space

The complete picture of infection dynamics generated by the SIS model requires the utilization of $2^N$ configuration vectors. For completeness sake, we discuss the algorithm to obtain the vector space using both string and numeric representations. Matrix elements of $\hat{T}$ in Equation (4) are calculated from an adjacency matrix and user input dictionary (lookup table) based on off-diagonal transition rules.

According to Equation (1), the configuration vector $|\mu\rangle$ is obtained from the binary representations of the labels $\mu$, as exemplified in Figure 5. There are two common equivalent routes to implement the configuration in computer codes. The first method employs string objects whereas the second method makes use of discrete mathematics. The second approach tends to be more efficient for two-state problems as optimized and native libraries for binary operations are widely available. For pedagogical purposes and generalization for more than two-states, we avoid exclusive binary operations in favor of usual discrete integer division and modulo operations.

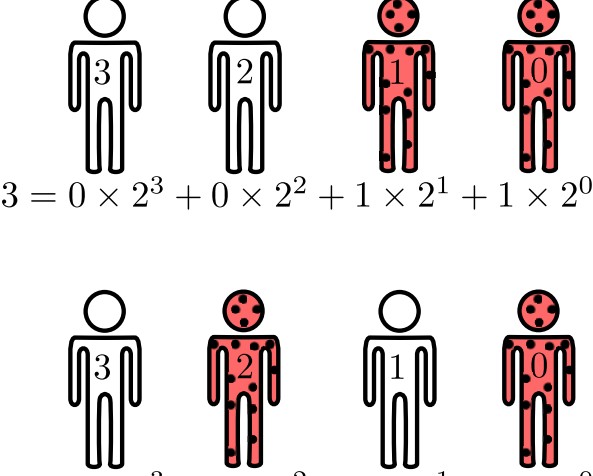

**Figure 5.** Agent configurations using binary representation for $\mu = 3$ and $5$ with $N = 4$. For $|\mu = 3\rangle = |0011\rangle$, whereas $|\mu = 5\rangle = |0101\rangle$. In both configurations, two agents are infected (red dotted).

In Python programming language, classes provide a convenient mechanism to enable both formalisms for each instanced object (vector). Here, the custom class SymConf is used to encapsulate two instance variables: *label* stores the string representation of $N$ agents, while *label_int* stores the corresponding integer number. In addition, the custom class also encapsulates three global class variables, *base*, *dimension*, and *basemax*, whose default values are 2, $N$, and $2^N$. Base corresponds to the number of available states per agent. The class main method generates the eigenvectors $|\mu_p\rangle$ with eigenvalue $\exp(-2i\pi p/N)$, relative to the cyclic permutation operator $\hat{P}$ using Equation (5).

In what follows, we address four relevant points regarding the permutation eigenvectors $|\mu_p\rangle$, namely, the criteria used to label eigenvectors; normalization; number of infected agents; and the permutation operation.

*Labels.* Equation (5) claims permutation eigenvectors are a linear combination of all configuration vectors related by cyclic permutations. Here, we set the convention to adopt the smallest value $\mu$ present in the linear combination to label the representative vector. As examples, consider the following representatives of $\mu = 1$, $N = 4$, and $p = 0, 1, 2, 3$:

$$|1_0\rangle = \frac{|0001 = 1\rangle + |0010 = 2\rangle + |0100 = 4\rangle + |1000 = 8\rangle}{\sqrt{4}}. \tag{6a}$$

$$|1_1\rangle = \frac{|1\rangle + i|2\rangle - |4\rangle - i|8\rangle}{\sqrt{4}}. \tag{6b}$$

$$|1_2\rangle = \frac{|1\rangle - |2\rangle + |4\rangle - |8\rangle}{\sqrt{4}}. \tag{6c}$$

$$|1_3\rangle = \frac{|1\rangle - i|2\rangle - |4\rangle + i|8\rangle}{\sqrt{4}}. \tag{6d}$$

The order convention is necessary to calculate the relative phase between configurations related by permutations in non-trivial linear combinations. For instance, consider the vector $|\phi\rangle = \hat{P}|\mu_p\rangle = (1/\mathcal{N}_\mu)\sum_k (e^{2i\pi p/N}\hat{P})^k \hat{P}|\mu\rangle$. Since $|\mu_p\rangle$ and $|\phi\rangle$ are related by a single cyclic permutation, they differ by a phase factor: $|\phi\rangle = e^{-2i\pi p/N}|\mu_p\rangle$. Note that the linear combination $\hat{P}|\mu_p\rangle + |\mu_p\rangle = (1 + e^{-2i\pi p/N})|\mu_p\rangle$ vanishes for $p = N/2$. Despite the simplicity of the previous example, it already illustrates the relevance of phase difference among cyclic vectors.

*Normalization.* According to Equation (5), the squared norm of representative vectors is:

$$\langle\mu_p|\mu_p\rangle = \frac{1}{\mathcal{N}_\mu}\sum_{k=0}^{N-1} e^{-2i\pi pk/N}\langle\mu|\hat{P}^{-k}|\mu_p\rangle = \frac{N}{\mathcal{N}_\mu}\langle\mu|\mu_p\rangle. \tag{7}$$

The evaluation of the scalar product $\langle\mu|\mu_p\rangle$ follows directly from Equation (5). One notices the configuration $|\mu\rangle$ may appear only once for several linear combinations $|\mu_p\rangle$, so that $\langle\mu|\mu_p\rangle = 1/\mathcal{N}_\mu$. For instance, this is the case of $\langle 1|1_p\rangle$. However, a given configuration $|\mu\rangle$ may contribute more than once if there exists an integer $1 \leq r \leq N$ such that $\hat{P}^r|\mu\rangle = |\mu\rangle$, i.e., after $r$ cyclic permutations the configuration repeats itself. Since $\hat{P}^N = \mathbb{1}$, it follows $N/r$ is the number of times the configuration $|\mu\rangle$ appears in $|\mu_p\rangle$. Each contribution adds $e^{2i\pi pm/N}/\mathcal{N}_\mu$ ($m = 0, 1, \ldots, N/r - 1$) in Equation (7). This result is conveniently summarized using the repetition number:

$$R_{\mu,p} = \sum_{m=0}^{N/r-1}{}' (e^{2i\pi p r/N})^m, \tag{8}$$

where the primed sum indicates $N/r$ in the upper limit is an integer number. Therefore, $\langle\mu|\mu_p\rangle = R_{\mu,p}/\mathcal{N}_\mu$ and one obtains $\mathcal{N}_\mu = \sqrt{NR_{\mu,p}}$ from Equation (7).

We now show two examples to consolidate the discussion around $R_{\mu,p}$ and $\mathcal{N}_\mu$, for $N = 4$ and two infected agents. The configuration state $|3\rangle = |0011\rangle$ requires $N$ cyclic permutations to repeat itself, so that $R_{3,p} = 1$ for any $p$ and the corresponding normalization for $|3_p\rangle$ is simply $\mathcal{N}_3 = \sqrt{N}$, as

expected. The first non-trivial case arises for $|5_p\rangle$ because the base configuration $|5\rangle = |0101\rangle$ satisfies $\hat{P}^2|5\rangle = |5\rangle$. According to Equation (8), $R_{5,p} = 1 + e^{4i\pi p/N}$ and assume only values: $R_{5,0} = R_{5,2} = 2$ and $R_{5,1} = R_{5,3} = 0$. Thus, depending on $p$, certain linear combinations are *forbidden* because they produce vectors with null norm, ensuring the correct dimension of vector space. The remaining non-null states for $N = 4$ are shown in Table 1 for further reference.

**Table 1.** Cyclic permutation eigenvectors with $N = 4$ agents. The first column shows the number of infected agents in the eigenvector. Each remaining column corresponds to a permutation sector $p$, and each row the corresponding state $|\mu_p\rangle$. The cross symbol indicates null-normed vector and the dimension of the vector space is $d = 2^4$.

| $n$ | $p = 0$ | $p = 1$ | $p = 2$ | $p = 3$ |
|---|---|---|---|---|
| 0 | $|0_0\rangle$ | $\times$ | $\times$ | $\times$ |
| 1 | $|1_0\rangle$ | $|1_1\rangle$ | $|1_2\rangle$ | $|1_3\rangle$ |
| 2 | $|3_0\rangle$ | $|3_1\rangle$ | $|3_2\rangle$ | $|3_3\rangle$ |
| 2 | $|5_0\rangle$ | $\times$ | $|5_2\rangle$ | $\times$ |
| 3 | $|7_0\rangle$ | $|7_1\rangle$ | $|7_2\rangle$ | $|7_3\rangle$ |
| 4 | $|15_0\rangle$ | $\times$ | $\times$ | $\times$ |

*Number of Infected Agents.* The number of infected agents using representative vectors is calculated as:

$$\langle\hat{n}\rangle_\mu = \sum_k \langle\mu_p|\hat{n}_k|\mu_p\rangle. \tag{9}$$

In the string representation, native string methods, such as *count('x')*, count the number agents with health state $x = 0, 1, 2 \dots$. If native methods are unavailable, one may always perform a comparative loop over the string. Algorithm A2. explains the standard procedure to count bits in the integer representation. It is worth mentioning that the operator $\sum_k \hat{n}_k$ commutes with $\hat{P}$.

*Permutation.* Cyclic permutations are the core transformations here. In the string representation, cyclic permutations consist of one copy and one concatenation call, as exemplified in Figure 6a.

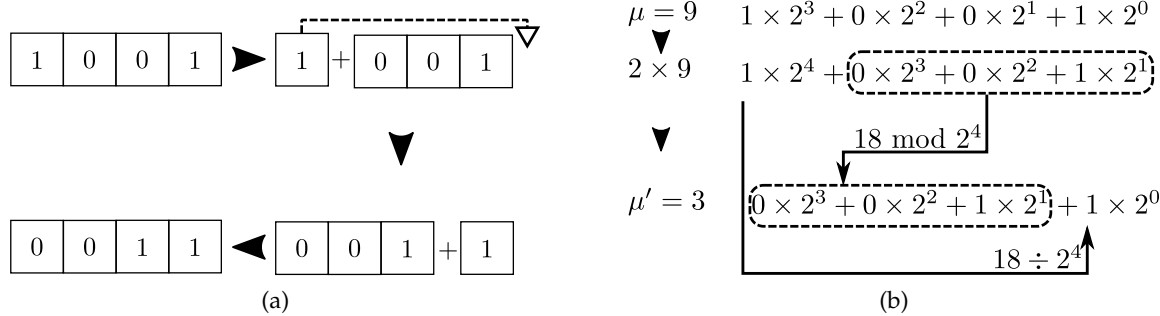

**Figure 6.** Cyclic permutation for configuration $\mu = 9$ with $N = 4$. (**a**) String representation executes one copy and one concatenation operation; (**b**) integer representation requires both integer division and modulo operation by $2^N$.

Meanwhile, in the integer representation, cyclic permutations are obtained using modulo and integer division: $\mu' = (2\mu \% 2^N) + (2\mu//2^N)$, the new configuration $\mu'$ is obtained from configuration $\mu$ taking the modulo of $2\mu$ by $2^N$ in addition to the result of the integer division $2\mu$ by $2^N$. Multiplication by the number of available states translates bit fields to the left. The modulo operation crops contributions larger than those available to $N$-bit fields. Integer division $2\mu/2^N$ selects the bit associated to largest binary position and shifts it to the lowest binary position (see Figure 6b).

Next, we focus our attention on the sector with $p = 0$, which plays an important role in epidemic models (see Section 5 for further discussion). This invariant subspace holds only symmetric linear combinations of configuration vectors. Incidentally, that also means that configurations with short

cycles—or large repetition numbers—can only have representative vectors with non-vanishing norms iff $p = 0$. The most important cases are: a) the all-infected configuration $|111 \cdots 1\rangle$, and b) disease-free configuration $|000 \cdots 0\rangle$. This occurs because these two configurations are invariant by every cyclic permutation available, including a single cyclic permutation (short cycle). As a direct consequence, the probability of disease eradication, $\pi_0(t)$, and the probability that the disease has infected each element of the population, $\pi_{2^N-1}(t)$, can only be evaluated at $p = 0$. Moreover, this sector holds the largest dimension being the worst scenario for numerical computations.

To construct the vectors for this particular sector, consider each integer $\mu$ in $[0, 2^N)$ as a potential candidate to assemble the symmetric vector spaces for fixed $p$. By performing $N - 1$ cyclic permutations over $|\mu\rangle$, one determines the representative state $|\mu_p\rangle$ in Equation (5), as well as the number of repetitions $R_{\mu,p}$, hence the norm $\mathcal{N}_\mu$. Algorithm A3. calculates the representative vector $|\mu_p\rangle$ associated with configuration $|\mu\rangle$. Due to the order convention adopted here, the string representation must be converted to the integer representation at the *if*-clause test. The representative configurations are then stored either in a list or dictionary. As an additional benefit, since vector spaces are independent of the problem at hand, the set of representatives may also be stored in a database for further use in different problems, as long as they are subjected to the same symmetry.

## 4. Matrix Elements

The next step is the evaluation of the transition matrix in the sector $p = 0$. Infection and recovery dynamics are the main actors in this context, as they inform the way representative vectors $|\mu_0\rangle$ interact with each other, $\hat{T}|\mu_0\rangle = \sum'_{\{\nu\}} T_{\nu\mu}|\nu_0\rangle$. The prime indicates the sum runs over all eigenvectors in the $p = 0$ sector, while cyclic permutation invariance implies:

$$\hat{T}|\mu_0\rangle = \frac{1}{\mathcal{N}_\mu} \sum_{k=0}^{N-1} \hat{P}^k \hat{T}|\mu\rangle \ . \tag{10}$$

Equation (10) tells us the action of $\hat{T}$ on the linear combination $|\mu_0\rangle$ is calculated from the simpler operation $\hat{T}|\mu\rangle$. The resulting vectors are then permuted, producing the corresponding matrix elements. The practical advantages of this method come from the order of the operations: By doing the transitions first and then finding the respective representatives, one divides the workload by a factor $N$. If the normal ordering were used instead, one would evaluate the transitions for each element of the linear combination and then find the corresponding representative, hence $N$ times the number of operations required with transition first. For instance, consider $\hat{T}|7_0\rangle$ for $N = 3$:

$$\begin{aligned}
\hat{T}|7_0\rangle &= \frac{1}{\mathcal{N}_7} \sum_{k=0}^{2} \hat{P}^k \hat{T}|7\rangle = \frac{\gamma}{\mathcal{N}_7} \sum_{k=0}^{2} \hat{P}^k \left(|3\rangle + |5\rangle + |6\rangle\right) \\
&= \frac{\gamma}{\mathcal{N}_7} \sum_{k=0}^{2} \hat{P}^k \left(|3\rangle + \hat{P}|3\rangle + \hat{P}^2|3\rangle\right) = \left(3\gamma \frac{\mathcal{N}_3}{\mathcal{N}_7}\right)|3_0\rangle \\
&= \sqrt{3}\gamma|3_0\rangle.
\end{aligned} \tag{11}$$

The relevant data structure for $\hat{T}$ are the off-diagonal transitions, which are further subdivided into two categories: one or two-body contributions. This is illustrated in Figure 7 for the SIS model. The finite set of transition rules are passed as a lookup table or, if available, a dictionary. Data is organized as follows: Each entry represents a one or two-body configuration whose value corresponds to one tuple. Each tuple holds two immutable values: the configuration to which the entry transitions to and the assigned coupling strength.

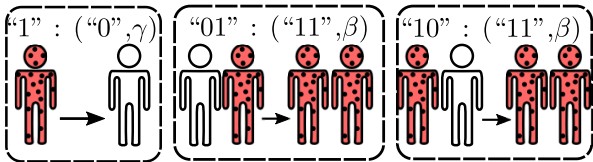

**Figure 7.** Off-diagonal transitions in the SIS model. Data structure follows the income-outcome convention. Data entries represent the current one-body (two-body) health state, whereas the corresponding data values, organized as tuples, express the outcome one-body (two-body) configuration and coupling strength.

With off-diagonal transition rules in hand, one-body actions are evaluated by scanning each agent and applying the corresponding transition rule in Algorithm A4.. The resulting one-body transitions are stored in the *outcome* variable. Figure 8 depicts an example for $N = 3$ and one infected agent at $k = 1$. Two-body operators differ from their one-body counterparts due to the fact they require two agent loops and information from the adjacency matrix $A$, as seen in Algorithm A5.. Figure 9 exhibits an example for $N = 3$. After both one- and two-body transitions are computed, the diagonal element is obtained *via* probability conservation: $T_{\mu\mu} = 1 - \sum'_{\mu \neq \nu} T_{\mu\nu}$. The process is iterated until all eigenvectors and their respective transitions are accounted.

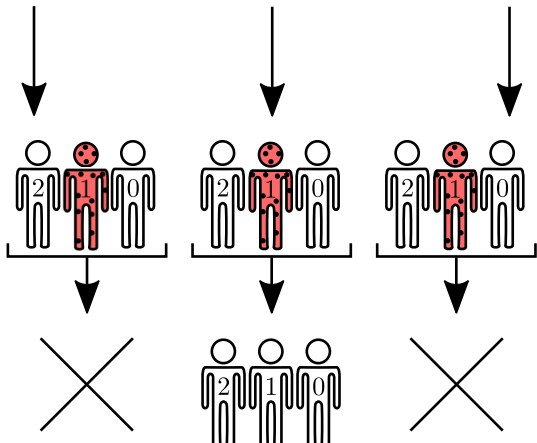

**Figure 8.** Recovery operator action on configuration vector $|2\rangle$, in the SIS model with $N = 3$. Non-vanishing transition is observed only for agent $k = 1$, which is infected, producing $|0\rangle$.

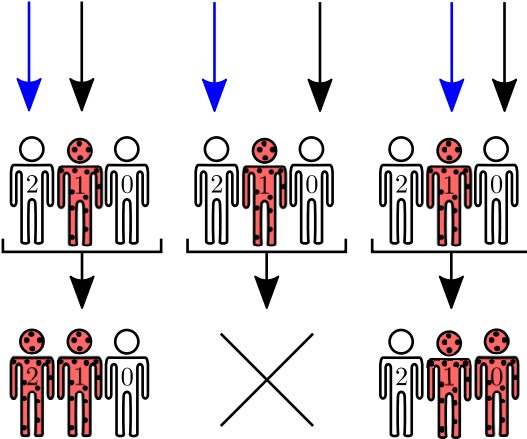

**Figure 9.** Infection operator action on configuration vector $|2\rangle$, in the SIS model with $N = 3$ and mean field network. Disease transmission events are evaluated for each pair of agents. Whenever the pair health state differs, and the pair also shares one connection expressed by the adjacency matrix, the configuration changes to contemplate the recently infected individual. For $|2\rangle$, $k = 1$ agent contaminates $k = 0$ ($k = 2$) agent, producing the configuration $|3\rangle$ ( $|6\rangle$ ).

## 5. Casimir Vector Space

The recent advances in the disease spreading dynamics in realistic populations are intimately linked to network theory [12,24]. Networks are traditionally associated with graphs holding a large number of nodes and links [25]. The graph must be large enough to produce a degree distribution, which describes the probability distribution of links per node. The degree distribution, or alternatively its statistical moments, characterizes the network type and its properties. However, some networks, including random networks, require an ensemble of graphs to provide an accurate picture. Thus, a graph becomes a sample or realization of the network. Statistical properties of networks are derived for each graph, followed by ensemble average and deviation. In practice, when graphs in the ensemble are large enough ($N \gg 1$) and representatives, statistics may also be evaluated for each graph and extrapolated as those of the network.

Two cases hold particular importance for applications of network theory in epidemic models: the mean field and random networks. In the first case, all agents are connected, meaning one infected agent may potentially infect anyone. Hence, the disease tends to spread faster than in constrained networks. Furthermore, all graphs in the mean field ensemble share the same adjacency $A^{\text{MF}}$. In the other case, the connection between agent $i$ and $j$ occurs with fixed probability $\rho$. However, graphs in the random network ensemble differ from each other. Here, we only consider ensemble averages as a way to extract statistical properties, which is equivalent to set $A_{ij}^{\text{random}} = \rho (1 - \delta_{ij}) = \rho A_{ij}^{\text{MF}}$. Thus, all relevant symmetries lie only in the mean field adjacency matrix $A^{\text{MF}}$. Naturally, $A^{\text{MF}}$ remains invariant under cyclic permutations, enabling the application of the algorithm explained in the previous sections. However, $A^{\text{MF}}$ is also symmetric under the action of any permutation, which drastically reduces the diagonal blocks of $\hat{T}$ from $O(2^N/N)$ to $O(N)$.

Here, our primary concern is to employ the cyclic permutation eigenvectors $|\mu_p\rangle$ to generate the eigenvectors of the complete permutation group, $|s, m; p\rangle$. The eigenvectors $|s, m; p\rangle$ reduce $\hat{T}$ in mean field or random networks to block diagonal form with dimension $O(N)$. The indices $s$ and $m$ may assume the following values $s = N/2, N/2 - 1, \ldots$ with $s > 0$ and $m = -s, -s + 1, \ldots, s$, respectively. The relationship between $s$ and $m$ are the same as those observed for quantum spin operators. The explanation goes as follows. As shown in Reference [16], Equation (4), in either mean field or random networks, contains operators $\hat{S}^{\pm} \equiv \sum_k \hat{\sigma}_k^{\pm}$ and $\hat{n} \equiv \sum_k \hat{n}_k$. From the important relation $\hat{n} = \hat{S}^z + N/2$, one retains spin operators and the upper bound $s = N/2$, as expected from the combination of $N$ $1/2$-spin particles.

In what follows, we only consider the $p = 0$ sector. First, let $\hat{S}^2 = (\hat{S}^z)^2 + (\hat{S}^+\hat{S}^- + \hat{S}^-\hat{S}^+)/2$ be the Casimir operator, so that $[\hat{S}^2, \hat{S}^\alpha] = [\hat{P}, \hat{S}^\alpha] = 0$ for $\alpha = z, \pm$ and $\hat{S}^2|s, m; 0\rangle = s(s + 1)|s, m; 0\rangle$. Accordingly, $[\hat{S}^2, \hat{T}] = 0$ and $s$ and $p$ are good quantum numbers. In general, the eigenvector $|s, m; p\rangle$ may always be expressed as:

$$|s, m; p\rangle = \sum_\mu c_\mu^{smp}|\mu\rangle . \tag{12}$$

Clearly, $c_\mu^{smp} = 0$ if the number of infected agents in the configuration $\mu$, $n_\mu = \sum_k \langle\mu_0|\hat{n}_k|\mu_0\rangle$, fails to satisfy the constraint $n_\mu = m + N/2$. The idea is to write Equation (12) as a linear combination of representative vectors $|\mu_p\rangle$ with $m + N/2$ infected agents, ensuring all available permutations are accounted for. The implications for numerical codes is quite obvious: it allows the reuse of numerical codes to obtain eigenvectors $|\mu_p\rangle$.

The most relevant sector for epidemic models contains the configuration with all (none) infected agents. According to previous sections, this implies $p = 0$ while $m = \pm N/2$ requires $s = N/2$. In the $(s = N/2, p = 0)$ sector, the desired linear combination is:

$$|s = N/2, m, p = 0\rangle = \frac{1}{\mathcal{N}} \sum_{\{\mu\}}' R_{\mu,0}^{-1/2}|\mu_0\rangle , \tag{13}$$

with normalization $|\mathcal{N}|^2 = \sum_\mu' |R_{\mu,0}|^{-1}$. The prime indicates the sum is subjected to the constraint $n_\mu = m + N/2$ for $m = -N/2, \dots, N/2$. The result in Equation (13) agrees with the standard theory of spin addition. Generalization for $p$ and $s$ is straightforward and omitted. It is worth mentioning the formalism adopted here already accounted for forbidden states in $p \neq 0$ sectors.

Examples are available to appreciate Equation (13) for increasing values of $N$. We begin considering $N = 4$. This translates into $s = 2$ and $m = -2, \dots, 2$. The relevant representative eigenvectors $|\mu_0\rangle$ are expressed in Table 2. The only non-trivial correspondence occurs for $m = 0$,

$$|2, 0; 0\rangle = \frac{\sqrt{2}|3_0\rangle + |5_0\rangle}{\sqrt{3}} = \frac{|0011\rangle + |1001\rangle + |1100\rangle + |0110\rangle + |0101\rangle + |1010\rangle}{\sqrt{6}}. \tag{14}$$

Next, consider $N = 6$ which fixes $s = 3$ and $m = -3, \dots, 3$. The eigenvector $|3, 0; 0\rangle$ holds contributions from four cyclic eigenvectors or, equivalently, 20 configurations:

$$\begin{aligned}
|3, 0; 0\rangle &= \frac{\sqrt{3}|7_0\rangle + \sqrt{3}|11_0\rangle + \sqrt{3}|19_0\rangle + |21_0\rangle}{\sqrt{10}} \\
&= \frac{|000111\rangle + |100011\rangle + |110001\rangle + |111000\rangle + |011100\rangle + |001110\rangle}{\sqrt{20}}. \\
&+ \frac{|001011\rangle + |100101\rangle + |110010\rangle + |011001\rangle + |101100\rangle + |010110\rangle}{\sqrt{20}}. \\
&+ \frac{|010011\rangle + |101001\rangle + |110100\rangle + |011010\rangle + |001101\rangle + |100110\rangle}{\sqrt{20}}. \\
&+ \frac{|010101\rangle + |101010\rangle}{\sqrt{20}}.
\end{aligned} \tag{15}$$

**Table 2.** Eigenvectors $|\mu_0\rangle$ with $N = 4$.

| $\mu_0$ | $R_{\mu,0}$ | $m$ | $|\mu\rangle$ |
|---|---|---|---|
| 0 | 4 | $-2$ | $|0000\rangle$ |
| 1 | 1 | $-1$ | $|0001\rangle$ |
| 3 | 1 | 0 | $|0011\rangle$ |
| 5 | 2 | 0 | $|0101\rangle$ |
| 7 | 1 | 1 | $|0111\rangle$ |
| 15 | 4 | 2 | $|1111\rangle$ |

## 6. Discussion

The algorithms presented in this study assumed only two health states for each agent. Generalization for $q$ number of states is readily available by changing to the integer representation $\mu = a_{N-1}q^{N-1} + \cdots + a_0 q^0$, with $a_k = 0, 1, \ldots, q-1$, concomitant with additional off-diagonal transitions. For instance, the susceptible-infected-recovered-susceptible (SIRS) ABEM generalizes the SIS model as it introduces the removed (R) health state for agents. This additional state often means the agent has recovered from the illness and developed immunity, has been vaccinated, or has passed away. In any case, once removed, the agent takes no part in the dynamics of disease transmission, hindering infection events [12]. As such, recovery with immunization or death events produce the transition $I \to R$, with probability $\gamma$ while vaccination $S \to R$ occurs with probability $\xi$. If death events are excluded, temporary immunization is achieved via $R \to S$ with probability $\eta$. Therefore, the vectors $|n_{N-1} \cdots n_0\rangle$ with $n_k = 0, 1$ or 2 describe configurations of the SIRS model. However, the algorithm to explore cyclic permutations remains unchanged as it explores symmetries of the underlying network. As a result, eigenvalues and number of sectors are the same, but degeneracy and eigenvectors change to accommodate the increased number of health states.

Parallelism merits further discussion. The computation of representative vector space may be performed in parallel by dividing the set of $q^N$ integers among $Q$ processes. Each process runs one local set of representative vectors which, posteriorly, is compared against the sets from the remaining processes. The union of all $Q$ sets produces the desired representative vector space. Parallelism is also obtained at the evaluation of $\hat{T}$: Columns ($|\mu_p\rangle$) are distributed among $Q$ processes and the corresponding matrix elements are calculated for each process. The union of all matrix elements from each process produces the complete description of $\hat{T}$ in the representative vector space. Lastly, parallelism is also available for sparse products $\hat{T}|\pi(t)\rangle$ necessary to execute the time evolution.

We also emphasize the algorithms explained here are most useful to evaluate quantities within a single permutation sector of $\hat{T}$. This is likely the case whenever the probability for disease eradication or complete population contamination are concerned. Another relevant situation occurs when the initial condition itself falls within a single sector. For instance, the initial probability vector $|\pi(0)\rangle = (1/3)(|001\rangle + |010\rangle + |100\rangle)$ states only one among $N = 3$ agents is infected. However, the identity of the infected agent is unknown a priori, so that configurations with one infected agent occurs with equal probability $1/N$. Now, the decomposition of $|\pi(0)\rangle$ in the $|\mu_p\rangle$ basis results in $|\pi(0)\rangle = (1/\sqrt{3})|1_0\rangle$. Thus, the time evolution of $|\pi(0)\rangle$ by the action of $\hat{T}$ is again restricted to a single permutation sector.

Without loss of generality, the initial condition can always be written as $|\pi(0)\rangle = \sum'_{\{\mu\}} \sum_{k=0}^{N-1} \pi_{\mu k} \hat{P}^k |\mu\rangle$, where the primed sum runs only over the indices $\mu$, which also labels the representative vectors. The cyclic permutation $\hat{P}^k$ generates the remaining configurations related to $|\mu\rangle$ whereas the coefficients $\pi_{\mu k}$ are the corresponding initial probabilities. Using the eigenvalue equation for $\hat{P}$, one calculates the scalar product:

$$\langle \nu_p | \pi(0)\rangle = \sideset{}{'}\sum_{\{\mu\}} \sum_{k=0}^{N-1} \pi_{\mu k} \langle \nu_p | \hat{P}^k | \mu\rangle = \sideset{}{'}\sum_{\{\mu\}} \sum_{k=0}^{N-1} \pi_{\mu k} e^{2i\pi pk/N} \langle \nu_p | \mu\rangle = \sqrt{N} \tilde{\pi}_{\nu p} \frac{R_{\nu p}}{\mathcal{N}_\mu}, \tag{16}$$

where $\tilde{\pi}_{\mu p} = N^{-1/2} \sum_k \pi_{\mu k} e^{2i\pi pk/N}$ is the discrete Fourier transform of $\pi_{\mu k}$. Using the previous example, with one infected among $N = 3$ agents, $|\pi(0)\rangle = \sum_{k=0}^{2} \pi_{0k} \hat{P}^k |0\rangle + \sum_{k=0}^{2} \pi_{1k} \hat{P}^k |1\rangle + \sum_{k=0}^{2} \pi_{3k} \hat{P}^k |3\rangle + \sum_{k=0}^{2} \pi_{7k} \hat{P}^k |7\rangle$, with $\pi_{\mu k} = \delta_{\mu 1}/3$ so that $R_{1p} = 1$, $\tilde{\pi}_{1p} = \delta_{p0}/\sqrt{3}$, and the previous result is recovered.

Now we address the case where the evaluation of the desired statistics requires several permutation sectors. In the worst case scenario, every permutation sector contributes equally to the computation. Therefore one must diagonalize each block in order to obtain the relevant eigenvalues and eigenvectors. As a crude approximation, one may consider that the $N$ blocks have the same dimension $d/N$ for a $d$-dimensional vector space. The complexity of diagonalization methods in the LAPACK library range from $O((d/N)^2)$ up to $O((d/N)^3)$ for each block [26], whereas the complexity range for full diagonalization is $[O(d^2), O(d^3)]$. Thus diagonalization of $N$ blocks reduces the total complexity from $N^{-1}$ up to $N^{-2}$. More importantly, blocks can be diagonalized in different processors because they are disjointed.

The algorithms presented here are most suitable for networks with invariance by cyclic permutations. However, they are also convenient whenever the algebraic commutator can be approximated by $[\hat{T}, \hat{P}] = \hat{O}$, where the operator $\hat{O}$ is symmetric under cyclic permutations, $[\hat{O}, \hat{P}] = 0$. In particular, $\hat{O} = q_0 \mathbb{1} + q_1 \hat{P}^y + \sum_{\beta=z,\pm} q_\beta \hat{S}^\beta$, with constant $q_j$ ($j = 0, 1, z, \pm$) and $y \in \mathbb{R}$, creates interesting disease-spreading dynamics, such as a localized disease source for $q_\beta = q\delta_{\beta,0}$.

Finally, we compare performances of the SIS ABEM using the transition matrix method with and without our algorithm. Numerical experiments were performed using Python on an Intel-PC i7-7700 3.8 GHz. The decision to pick up Python instead of a more performance-oriented language was based on the ability to quickly disseminate the method. For data intensive research, we strongly recommend performance-oriented languages, such as C or high-performance Fortran. The results are summarized in Table 3. As expected, cyclic permutations greatly improve computation times, most noticeable for large populations sizes. For $N = 20$, the improved numerical code runs two orders of magnitude faster, while only consuming a fraction—about 6%—of the original memory. We reiterate methods involving the transition matrix to compute the probabilities of each configuration available to the system $\pi_\mu(t)$, with $\mu = 0, 1, \ldots, 2^N - 1$, up to numerical errors (floating point and rounding errors), often around $O(10^{-12})$. Because they include all configurations, they can provide accurate statistics and data predictions along the evolution of the epidemics. However, direct Monte Carlo methods (DMCM) are far more efficient if one is solely interested in a few statistical moments of relevant variables, not in the entire joint pdf [27,28]. There are mainly two flavors of DMCM, depending on whether the time interval is fixed or distributed according to a given PDF [29]. The latter case is more commonly known as the Gillespie algorithm [30–32], and it has been successful to simulate epidemics. In DMCMs, execution times are directly related to the number of independent runs $m$, with error scaling as $m^{-1/2}$. Usually, $m \sim O(10^6)$ produces errors around $O(10^{-3})$. Smaller errors can be obtained by increasing $m$. Regardless, DMCM are always more efficient if the joint PDF is not required, as they probe the configurations that are more likely to occur. Indeed, computation times of DMCM with $N = 20$, $\tau = 0.19$ s, are far lower than the 41 s obtained previously. Furthermore, DMCM hold small memory footprint and can simulate ABEM with $N \sim O(10^4)$.

**Table 3.** Computation times and memory usage of time evolution of the SIS agent-based epidemic models (ABEM) for various population sizes, with (block) and without (full matrix) cyclic permutations.

| N | Time (s) | | Memory (MB) | |
|---|---|---|---|---|
| | **Block** | **Full Matrix** | **Block** | **Full Matrix** |
| 10 | 0.02(9) | 0.45(7) | 23.3(0) | 24.7(1) |
| 12 | 0.10(0) | 2.91(4) | 23.9(8) | 32.3(9) |
| 14 | 0.41(1) | 19.59(6) | 26.5(2) | 62.4(8) |
| 16 | 1.76(6) | 132.31(5) | 34.8(1) | 201.0(8) |
| 18 | 8.38(4) | 837.86(9) | 66.4(5) | 784.6(8) |
| 20 | 41.93(1) | 4529.09(7) | 184.0(4) | 3251.0(7) |

## 7. Conclusions

ABEM describe the stochastic dynamics of disease-spreading processes in networks. Direct investigation of epidemic Markov processes is often hindered due to the exponential increase of the dimension of the vector space with the number of agents. By exploiting cyclic permutation symmetries, relevant elements of the dynamics are confined to a single permutation sector, significantly reducing computational efforts. In practical terms, by selecting a single cyclic permutation eigensector, one selects only relevant information from the stochastic process. The $p = 0$ sector holds particular importance, as it contains configurations where none or all agents are infected, with dimension scaling as $O(N)$ for highly connected networks. Our findings show that using symmetric basis significantly improves computation times and reduces memory usage, providing a detailed picture of the joint probability distribution function. This development allows for a more detailed investigation of fluctuations and correlation functions in epidemics. For global statistics that describe the evolution of the epidemic, DMCM provide much faster computation times subjected to a given statistical error. In closing, the inclusion of finite symmetries brings down ABEM to the same footing of compartmental models regarding the number of agents but does not neglect the role played by fluctuations.

**Author Contributions:** A.S.M. and G.M.N. designed the research; A.C.P.M. and G.M.N. performed the research and wrote computational codes; A.C.P.M. verified numerical results; G.M.N. wrote the paper; G.C.C. and A.S.M. edited the paper. All authors reviewed the manuscript.

**Funding:** G.M.N. thanks CAPES/PNPD 88887.136416/2017-00, A.S.M. holds grants from CNPq 307948/2014-5, G.C.C. acknowledges funding from CAPES 067978/2014-01 and A.C.P.M. acknowledges grant CNPq 800585/2016-0.

**Conflicts of Interest:** The authors declare no conflict of interest.

## Abbreviations

The following abbreviations are used in this manuscript:

| | |
|---|---|
| ABEM | Agent-based epidemic model |
| NPDF | Network probability distribution function |
| SIS | Susceptible-infected-susceptible |
| SIRS | Susceptible-infected-recovered-susceptible |
| WHO | World health organization |

## Appendix A. Algorithms

*Appendix A.1. Time Evolution*

---
**Algorithm A1.** Time Evolution

---
**Require:** $p \in \mathbb{N}$, matrix $A$ and off-diagonal transitions

  $S = \{ \quad \}$                ▷ Basis

  **for** $\mu = 0$ to $\mu < 2^N$ **do**

    $\psi, \mathcal{N}_\psi \leftarrow$ calculates eigenvector and norm from $\mu$

    Add $\psi$ to $S$

  **end for**                ▷ $p$ invariant eigensector

  **for** $\psi$ in $S$ **do**

    **for** $k = 0$ to $k < N$ **do**

      $\psi' \leftarrow$ off-diagonal transitions from $k$-th component of $\psi$

      Evaluate $T_{\psi'\psi}$             ▷ Sparse storage

    **end for**

  **end for**

  $\pi \leftarrow$ initial condition

  **for** $t = 0$ to $t < t_{\max}$ **do**

    $\pi \leftarrow \hat{T} \times \pi$

  **end for**                ▷ End time evolution

---

*Appendix A.2. Number of Infected Agents*

---
**Algorithm A2.** Number of Infected Agents

---
1:  **function** COUNT($\mu$,count)

2:     c $\leftarrow \mu$

3:     count $\leftarrow 0$

4:     **for** $k = 0$ to $k < N$ **do**

5:       count $\leftarrow$ count + c % 2

6:       c $\leftarrow$ c // 2

7:     **end for**

8:  **end function**

---

*Appendix A.3. Representative Vectors*

---
**Algorithm A3.** Representative Vectors

---
1:  **function** REPRESENTATIVE($\mu, \psi, r$)

2:     $\psi \leftarrow \mu$

3:     $r \leftarrow 1$

4:     **for** $k = 0$ to $k < N - 1$ **do**

5:       $\mu \leftarrow \hat{P}\mu$

6:       **if** $\mu < \psi$ **then**

7:         $\psi \leftarrow \mu$

8:       **else if** $\mu = \psi$ **then**

9:         $r \leftarrow r + 1$

10:       **end if**

11:     **end for**

12: **end function**

---

*Appendix A.4. One-Body Off-Diagonal Transitions*

---

**Algorithm A4.** One-Body Off-Diagonal Transitions

---

1: **function** ONEBODY(label,rules,output)
2:　　**for** $k = 0$ to $k < N$ **do**　　　　　　　　　　　　　　　▷ Loop over agents
3:　　　　**if** label[k] in rules **then**
4:　　　　　　new ← label with label[k] ← rule[label[k]][0]
5:　　　　　　output[new] ← coupling
6:　　　　**end if**
7:　　**end for**
8: **end function**

---

*Appendix A.5. Two-Body Off-Diagonal Transitions*

---

**Algorithm A5.** Two-Body Off-Diagonal Transitions

---

1: **function** TWOBODY(L,A,rules,outcome)
2:　　**for** $j = 0$ to $j < N$ **do**
3:　　　　**for** $i = 0$ to $i < N$ **do**
4:　　　　　　$q \leftarrow (L_j L_i)$
5:　　　　　　**if** $q$ in rules **then**
6:　　　　　　　　$x \leftarrow L$
7:　　　　　　　　$x_j \leftarrow \text{rules}[q]_{00}$
8:　　　　　　　　$x_i \leftarrow \text{rules}[q]_{01}$
9:　　　　　　　　$\text{output}[x] \leftarrow \text{output}[x] + A_{ji}$
10:　　　　　　**end if**
11:　　　　**end for**
12:　　**end for**
13: **end function**

---

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
