# Peer review of "Finite Symmetries in Agent-Based Epidemic Models"

_mca, doi:10.3390/mca24020044_

Round 1
Reviewer 1 Report
Review for [MCA] Manuscript ID: mca-466100
The manuscript presents an approach for understanding the dynamics of diseases through finite symmetries in agent-based epidemic models. The authors focus on the SIS model and extend the comments to SIR and SIRS models. The proposed method optimizes the computational performance by reducing the computational costs, considering the symmetries in the transition matrix (which relates the events of either recovery or infection). The authors are quite didactic in explaining their method with examples and discussing their algorithms. Further, in the appendix, they provide detailed algorithms.
The manuscript is original, fits the journal’ scope and is of interest of the readers. It is well written and I recommend for publication after minor revisions as follows.
1) In the Introduction, the authors could include a review of ABEM models, such as:
Willem, L., Verelst, F., Bilcke, J., Hens, N., & Beutels, P. (2017). Lessons from a decade of individual-based models for infectious disease transmission: a systematic review (2006-2015). BMC infectious diseases, 17(1), 612. doi:10.1186/s12879-017-2699-8.
2) To make clear the recovery rate and the infection rate in figure 2, the authors could write over the arrows gamma and beta, respectively.
3) In line 34, it is confused to which agent the authors refer, namely, the third agent is the n_3, but in the sequence 1000 it is the fourth not the third.
4) In line 92, the authors could specify that Python is a programming language, such as “In Python programming language, …”.
5) Between lines 191 and 198, the authors comment about SIRS model. It is a bit confusing because they first consider the SIR model (lines 192-195) and at the end of the paragraph they consider the SIRS model. They could start with the SIRS model, which generalize the SIS model, as they propose. And later explain a particular case where the recovered agents are, in fact, removed (SIR model).
6) In the last phrase before equation (27), the word “equation” is repeated. (I don’t know why this set of the manuscript has no line number.)

Author Response
Dear Editor,
We have carefully read the Referees’ reports about the manuscript “Finite symmetries in agent-based epidemic models”. We sincerely thank them for their efforts and critical reading of the paper. The revised manuscript contains extensive amendments to address their critiques, comments and suggestions. In particular, we acknowledge their shared concern on the lack of evidence or proof that the algorithm explained in the paper produces any substantial improvement. In the revised manuscript, performance improvements are displayed in Table 3 to solve the issue. We believe this content addition is crucial for the reader to determine the scientific value of the paper. Moreover, we also acknowledge and thank the Referees for providing us with pertinent references.
The modifications are displayed in red in the revised manuscript to facilitate their identification throughout the text. We believe the corrections and suggestions put forward by the Referees improved the clarity of the paper. Several sentences have been re-written to improve wording and clarity, including the abstract. The complete list of amendments are detailed in the attached .pdf file.
Cordially,
The Authors
_____________________________________________________
Referee 1
In the Introduction, the authors could include a review of ABEM models
We thank the Referee by the reference. We cite it at Introduction.
To make clear the recovery rate and the infection rate in figure 2, the authors could
write over the arrows gamma and beta, respectively.
We have changed figure 2 to include the β and γ.
In line 34, it is confused to which agent the authors refer, namely, the third agent is
the n 3 , but in the sequence 1000 it is the fourth not the third.
We agree. In the revised manuscript, we now state the agent with k = 3 is infected in
line 38.
In line 92, the authors could specify that Python is a programming language, such as
“In Python programming language, ...”.
We agree with the comment. We have used the suggestion put forward by the Referee.
Between lines 191 and 198, the authors comment about SIRS model. It is a bit confusing because they first consider the SIR model (lines 192-195) and at the end of the paragraph they consider the SIRS model. They could start with the SIRS model, which generalize the SIS model, as they propose. And later explain a particular case where the recovered agents are, in fact, removed (SIR model).
To amend the issue, the revised manuscript only makes references to the SIRS model in the Discussion section. It simplifies the narrative and allows us to spend more
time explaining that our method to exploit permutation symmetries remains largely
unchanged.
In the last phrase before equation (27), the word “equation” is repeated. (I don’t know why this set of the manuscript has no line number.)
Following Referee’s 2 suggestions, we transformed several equations into inline equations. In the process, the “equation Eq.” passage was amended as well.

Reviewer 2 Report
In this paper, using agent-based epidemic models, an algorithm to explore permutation symmetries was presented. The following are some comments needing to be addressed:
1. The research background is not well explained. What is the motivation for the presented algorithm? The authors should give an explanation in the introduction.
2. There is no need for all equations to be numbered in this paper.
3. It would be better if the author could provide some proof.
4. The references are not new, and the authors should view more recently papers.
Author Response
Dear Editor,
We have carefully read the Referees’ reports about the manuscript “Finite symmetries in agent-based epidemic models”. We sincerely thank them for their efforts and critical reading of the paper. The revised manuscript contains extensive amendments to address their critiques, comments and suggestions. In particular, we acknowledge their shared concern on the lack of evidence or proof that the algorithm explained in the paper produces any substantial improvement. In the revised manuscript, performance improvements are displayed in Table 3 to solve the issue. We believe this content addition is crucial for the reader to determine the scientific value of the paper. Moreover, we also acknowledge and thank the Referees for providing us with pertinent references.
The modifications are displayed in red in the revised manuscript to facilitate their identification throughout the text. We believe the corrections and suggestions put forward by the Referees improved the clarity of the paper. Several sentences have been re-written to improve wording and clarity, including the abstract. The complete list of amendments are detailed in the attached .pdf file.
Cordially,
The Authors
______________________________________________
Referee 2
The research background is not well explained. What is the motivation for the presented algorithm? The authors should give an explanation in the introduction.
We agree. In line 42, at “The transition matrix (...) numerical methods.” we explain
the reason to consider the transition matrix, namely, the computation of the entire
probability distribution function. This is equivalent to solving the master equation in
discrete time, which allows one to calculate any statistics of the problem at any given
instant of time.
There is no need for all equations to be numbered in this paper.
We have reduced the amount of numbered equations.
It would be better if the author could provide some proof.
At line 281, an entire paragraph is directed to compare our method with he non-
symmetric version as well as direct Monte Carlo methods. The later one are much
more efficient to calculate statistical moments of a given pdf, such as density of infected agents. In contrast, methods based on the transition matrix provides access to the joint pdf. We believe this in an important discussion so that readers can evaluate
which method is more adequate for their own needs.
The references are not new, and the authors should view more recently papers.
We have added more recent references related to the subjects of the paper.

Reviewer 3 Report
In this paper, the authors introduced an algorithm to explore permutation symmetries for the well-known SIS epidemic model using an agent-based epidemic approach. The proposed algorithm restricts the stochastic process to a sector of the vector space, labeled by a single permutation eigenvalue. One main advantage of the proposed algorithm is that the transition matrix is reduced to a block diagonal form, which allows to reduce the computational time of the simulation process.
The article is interesting and could be useful for other scientists working in epidemic modeling. Some issues that need to be improved before publication are the following:
1- Some comments regarding the Gillespie algorithm that has been used to model similar epidemic processes and include reduction of computation time. How the proposed algorithm compares with some Gillespie algorithms.
2- Line 79: Expand the explanation of reducing T to a block diagonal form whenever [P,T]=0.
3- Line 126: The authors are working with the SIS model. However, they mentioned that sector p=0 holds all the infected and recovered vectors. Please elaborate.
4- Line 166: Explain why the network is an ensemble of graphs ? Could be viewed just as one graph ?
5- Line 192: What would be the main differences when using this algorithm for the SIRS ?
Finally, since the authors claim the advantages of the proposed algorithm it seems necessary to include simulation results for small and large N and compare with a classical algorithm for agent-based models. In particular, for the SIS model. Compare computation times, and observe the dynamics of infected agents.
Author Response
Dear Editor,
We have carefully read the Referees’ reports about the manuscript “Finite symmetries in agent-based epidemic models”. We sincerely thank them for their efforts and critical reading of the paper. The revised manuscript contains extensive amendments to address their critiques, comments and suggestions. In particular, we acknowledge their shared concern on the lack of evidence or proof that the algorithm explained in the paper produces any substantial improvement. In the revised manuscript, performance improvements are displayed in Table 3 to solve the issue. We believe this content addition is crucial for the reader to determine the scientific value of the paper. Moreover, we also acknowledge and thank the Referees for providing us with pertinent references.
The modifications are displayed in red in the revised manuscript to facilitate their identification throughout the text. We believe the corrections and suggestions put forward by the Referees improved the clarity of the paper. Several sentences have been re-written to improve wording and clarity, including the abstract. The complete list of amendments are detailed in the attached .pdf file.
Cordially,
The Authors
____________________________________________
Referee 3
Some comments regarding the Gillespie algorithm that has been used to model similar epidemic processes and include reduction of computation time. How the proposed algorithm compares with some Gillespie algorithms.
Transition matrix techniques are usually applied to unveil the entire joint probability
distribution function. This is often an computationally intensive task because the
system tends to scale exponentially with the system size. In our paper, we mitigate
the issue by breaking the vector space in a invariant subspaces (sectors). During
the review, it became clear the previous manuscript failed to convey that methods
involving transition matrix are often associated with obtaining the complete joint
probability distribution function. To address the issue, we extended our discussion at
line 42.
Whenever joint probability functions of the system are not required, we explicitly
state that direct Monte Carlo methods (DMCM) are more efficient than our method.
It includes a discussion about the Gillespie algorithm and DMCMs have been added
in line 292 at “However, direct Monte Carlo methods are far more efficient (...)”.
Execution times are included to back up the claim so the reader can compare both
techniques and their respective scopes.
Line 79: Expand the explanation of reducing T to a block diagonal form whenever [P,T]=0.
We improved the explanation by first extending the paragraph in line 74 at “Therefore, the trick to simplify (...) enhancing computation times”. Right after Eq. (5), we added the sentences “with p = 0, 1, . . . , N − 1 (...) eigenvalue sector”. Finally, at line 101 we expand the explanation and reasoning behind our claim.
Line 126: The authors are working with the SIS model. However, they mentioned that sector p=0 holds all the infected and recovered vectors. Please elaborate.
At line 156, we explain that the sector with p = 0 contains only symmetric configurations. The disease-free configuration and the one in which all agents are infected belong to this particular sector. It is, thus, the most relevant sector. And because transitions between different p-sectors are forbidden by symmetry arguments, it is possible to concentrate the analysis – if desired – to a single sector.
Line 166: Explain why the network is an ensemble of graphs ? Could be viewed just as one graph ?
The previous version of the manuscript was worded in a misleading manner. Networks are intuitive objects but their precise definition is somewhat odd. For instance, in his review, Newman states in that networks are graphs [Newman SCIAM Review 2003]. Later on his paper, when dealing with random networks, an ensemble of graphs is more suitable to describe random graphs. In line 192, at “Networks are traditionally (...) accurate picture.” we expand our discussion on the subject.
Line 192: What would be the main differences when using this algorithm for the SIRS?
In line 242 at “Therefore, the vectors (...) health states.” we explain that the most significant change occurs for configuration vectors. The procedure to extract representative vectors remains largely unchanged.
Finally, since the authors claim the advantages of the proposed algorithm it seems necessary to include simulation results for small and large N and compare with a classical algorithm for agent-based models. In particular, for the SIS model. Compare computation times, and observe the dynamics of infected agents.
At line 281, we introduce a new paragraph and a table with computation times. We
compare it with classical DMCM models, including the Gillespie algorithm, pointing
its weaknesses and advantages.

Round 2
Reviewer 2 Report
The revised article can be accepted.
Reviewer 3 Report
The authors improved the paper and now the ideas are clear.